# Diagnostic and Prognostic Roles of Procalcitonin and Other Tools in Community-Acquired Pneumonia: A Narrative Review

**DOI:** 10.3390/diagnostics13111869

**Published:** 2023-05-26

**Authors:** Sedat Ozbay, Mustafa Ayan, Orhan Ozsoy, Canan Akman, Ozgur Karcioglu

**Affiliations:** 1Department of Emergency Medicine, Sivas Numune Education and Research Hospital, Sivas 58040, Turkey; ilsedat58@hotmail.com (S.O.); drmustafayan@gmail.com (M.A.); rhnzsy@gmail.com (O.O.); 2Department of Emergency Medicine, Canakkale Onsekiz Mart University, Canakkale 17100, Turkey; drcananakman@gmail.com; 3Department of Emergency Medicine, University of Health Sciences, Taksim Education and Research Hospital, Beyoglu, Istanbul 34098, Turkey

**Keywords:** pneumonia, bacteremia, sepsis, procalcitonin, diagnosis, biomarker, outcome, antibiotic stewardship

## Abstract

Community-acquired pneumonia (CAP) is among the most common causes of death and one of the leading healthcare concerns worldwide. It can evolve into sepsis and septic shock, which have a high mortality rate, especially in critical patients and comorbidities. The definitions of sepsis were revised in the last decade as “life-threatening organ dysfunction caused by a dysregulated host response to infection”. Procalcitonin (PCT), C-reactive protein (CRP), and complete blood count, including white blood cells, are among the most commonly analyzed sepsis-specific biomarkers also used in pneumonia in a broad range of studies. It appears to be a reliable diagnostic tool to expedite care of these patients with severe infections in the acute setting. PCT was found to be superior to most other acute phase reactants and indicators, including CRP as a predictor of pneumonia, bacteremia, sepsis, and poor outcome, although conflicting results exist. In addition, PCT use is beneficial to judge timing for the cessation of antibiotic treatment in most severe infectious states. The clinicians should be aware of strengths and weaknesses of known and potential biomarkers in expedient recognition and management of severe infections. This manuscript is intended to present an overview of the definitions, complications, and outcomes of CAP and sepsis in adults, with special regard to PCT and other important markers.

## 1. Introduction

### 1.1. Community-Acquired Pneumonia (CAP): Definitions and Diagnostic Approaches

Pneumonia is the acute infectious process of the pulmonary parenchymal tissue. Community-acquired pneumonia (CAP) is the leading healthcare concern worldwide. CAP inflicts patients in the community, as opposed to nosocomial pneumonia [1,2]. Pneumonia is defined as symptoms and clinical manifestations such as high body temperature, cough, dyspnea, recent onset of weakness, and pleuritic chest pain [3].

#### The Situation in Turkey and the World

Different studies revealed various prevalence figures in pneumonia. The annual incidence of pneumonia is reported to vary between 0.28 and 1.16%, and both incidence and mortality increase, especially in advanced ages [4,5]. A population-based study in Germany showed that the yearly incidence of CAP admitted to the hospital was 0.3% [6]. Male patients are reported to be more likely to be admitted to the hospital than females [7].

Similar to the universal data, the incidence of pneumonia increases with age in Turkey. Pneumonia mortality varies between 1–60% in relation to the severity of the disease, and this rate rises significantly in hospitalized patients (10.3–60%) [8,9]. Patients with pneumonia accounted for 1.9% of all hospitalizations in Turkey [10].

In a population-based multicentric study in Turkey, pneumococcal pneumonia was diagnosed with at least one supportive laboratory finding for *Streptococcus pneumoniae* (blood or sputum culture or urinary antigens) in those with chest X-ray signs of pneumonia. Four hundred sixty-five patients were recorded to have CAP, and among them, 13% had pneumococcal pneumonia [11]. In another study focusing on elderly patients in Turkey, *Pseudomonas* spp. (26.6%) was the most common agent, followed by *Streptococcus pneumoniae* (23.3%) [12]. Other typical bacterial agents were *Staphylococcus aureus* (13.3%), *Haemophilus influenzae* (10%), and *Acinetobacter* spp. (10.0%). Around one-fourth of the *S. aureus* strains had methicillin resistance.

CAP is accompanied by significant morbidity, mortality, and healthcare expenses. The death rate in hospitalized patients with CAP is reported to range between 10 and 12% [13]. McLaughlin et al. criticized most studies in this area and postulated that most exclusion criteria (i.e., those hospitalized with CAP or immunocompromising conditions) or restrictions for defining cases (e.g., only including pneumonias coded in the primary diagnosis position) paved the way to systematic bias to underestimate CAP incidence [14]. In the other research without these restrictive criteria, only around 2% of the patients were hospitalized annually. In elderly patients, the percentage of hospitalization was found to lie between 847 and 3500 per 100,000 patients CAP annually.

Although CAP has a predilection for the elderly, newborns, or young infants, it is reported at all ages, especially those with compromised immunity and/or other comorbidities. Immune- and age-related changes make these subgroups more vulnerable to pneumonia. For example, the incidence of CAP in the elderly (>65 years of age) comprise around 3% inhabitants/year [15,16].

## 2. Clinical Findings

No clear group of symptoms and signs has been found to reliably predict whether or not the patient has the disease [17]. Certain clinical findings support the diagnosis of pneumonia. Signs and symptoms of CAP include cough, fever, sputum production, chest pain (mostly pleuritic in nature), dyspnea, crackles, and diminished or bronchial breath sounds, which may be encountered upon physical examination. Mucopurulent sputum production is most frequently detected in conjunction with pneumonia of bacterial origin, whereas watery sputum is suggestive of atypical pathogens. Nausea, vomiting, diarrhea, and mental status changes are also noted frequently. Chest pain is a major complaint in a third of cases, chills have been reported in up to half of cases, while rigors have been reported in 15% of cases [18]. An acutely ill patient with somatic chest pain and leukocytosis (>15 × 10^9^/L) is suggestive of *S. pneumoniae* aetiology [19].

## 3. The Diagnosis of CAP

The diagnosis of CAP generally necessitates an infiltration on CXR in a patient with fever, dyspnea, cough, and sputum.

While S. pneumonia is the most commonly isolated agent, *S. aureus*, *Haemophilus influenzae*, *Enterobacteriaceae*, *Legionella pneumophila*, *Mycoplasma pneumoniae*, and *Chlamydophila pneumonia* are among the culprits in patients with CAP. A Swedish study disclosed that in hospitalized patients with CAP, Pneumococci are the dominant agent, followed by *Haemophilus influenzae* and *Mycoplasma pneumonia* [20,21]. On the other hand, elderly patients have a different order of frequency of culprit agents in CAP (Table 1) [16,22]. Table 2 summarizes the differential diagnosis in patients presenting with cough.

### 3.1. Radiological Findings

#### 3.1.1. Chest X-rays

Chest X-rays (CXR, PA and lateral) can mostly be adequate for decision-making in suspected patients, which render CT scans not necessary in selected situations. The diagnosis of CAP is generally based on the presence of predefined clinical properties and is supported by simple imaging modalities, mostly by CXR [23]. In this regard, CAP presents as one of three patterns as follows:1.Focal nonsegmental or lobar pneumonia.2.Bronchopneumonia in multiple foci or lobular pneumonia.3.Patterns compatible with interstitial pneumonia (focal or diffuse).

False-negative CXR can be seen in the initial stage of pneumonia in some situations, including patients with neutropenia, dehydration, and immunocompromise. A special example is *Pneumocystis carinii* pneumonia (PCP, a.k.a. *Pneumocystis jirovecii* pneumonia), in which spiral CT scans can be needed to adequately visualize findings suggestive of infection.

High-resolution computed tomography (HRCT) usually demonstrates the pattern and distribution of pneumonia more accurately than the CXR [24,25]. It is not routinely ordered in the diagnosis of patients with suspected CAP because of cost-effectiveness principles. Instead, HRCT can be ordered as an adjunct to CXR in selected cases. For example, HRCT has been postulated to be a useful alternative to RT–PCR in the diagnosis of COVID pneumonia, in which a negative test can rule out the diagnosis of COVID pneumonia [26].

#### 3.1.2. Ultrasonography (USG)

Lung USG has been employed more commonly in the last decade to diagnose pneumonia with inappreciable diagnostic value in the patients *in extremis* who are hard to be transferred to the radiology unit. The sensitivity of lung USG was reported to be between 80 and 90%, and the specificity between 70 and 90% [27,28].

### 3.2. Microbiological Work-Up

#### 3.2.1. Sputum Gram Stain

A systematic review and Bayesian meta-analysis pointed out that a gram stain was adequately accurate to diagnose *S. pneumoniae* and *H. influenzae* in those with CAP [29]. With good-quality specimens, it can form a basis of clinical actions for specified antibiotic therapies for certain pathogens.

#### 3.2.2. Blood Count

An increased WBC count (up to 30,000/mm^3^) and a leftward shift are common findings, whereas leukopenia is suggestive of a poor outcome. The co-existence of fever, cough, tachycardia, and crackles had a sensitivity below 50% when CXR was used as a reference standard [30]. *Legionella* spp., Influenza A and B, MERS–CoV and SARS–CoV, and community-associated methicillin-resistant *Staphylococcus aureus* (CA-MRSA) are among these organisms.

#### 3.2.3. Blood Cultures

Blood cultures (BCs) and sputum gram stains and cultures should be obtained and studied in severe, hospitalized patients. BCs are expected to be positive in around one-fifth of patients. Patients with severe CAP requiring ICU admission, especially, should have BCs, *Legionella* and pneumococcus urinary antigen tests, and sputum culture. BCs are recommended in severe and critical patients with CAP because positive results indicate the specific microbial diagnosis in most cases [31]. False-positive BCs can be encountered in one-tenth of the patients [32]. Studies pointed out that positive BCs rarely result in a change of antibiotic treatment regimens [33].

#### 3.2.4. Molecular Methods

The potential advantages of molecular methods are speed and enhanced sensitivity and specificity [34,35]. These methods are available in most centers to elucidate viral agents and some bacteriae, including *M. pneumoniae* and *Legionella pneumophila.* Polymerase chain reaction (PCR) boosts the accuracy of the microbiological tests for patients with CAP with its rapid turnaround time [20,36]. Since PCR specimens can be contaminated by the airway flora, a quantitative or semiquantitative PCR assay is needed in most cases [20,37,38].

### 3.3. Biomarkers

#### 3.3.1. Lactate

Lactate is another biomarker with diagnostic and prognostic value in most severe infections [39]. Research disclosed that lactate was able to predict poor outcomes in CAP patients in the acute setting and augmented the predictive power for death [40]. High lactate value is associated with mortality of up to one-third of the samples in patients with CAP. An elevated lactate level suggests hypoperfusion and a marker for grave clinical course [41,42]. In accordance with the updated criteria for sepsis, both hypotension, which prompted inotropic infusions, and high lactate (>2 mmol/L) are necessary for the recognition of septic shock [43].

A recent study analyzed the impact of adding lactate levels to the Rapid Emergency Medicine Score (REMS) system to predict death and prognosis in the middle-aged and elderly (>40 years of age), who were admitted to the ED with dyspnea [44]. The REMS + L score (*p* < 0.001) was found to be more accurate than REMS (*p* < 0.001) and lactate values (*p* < 0.001) in the prediction of death.

#### 3.3.2. Monocyte Human Leukocyte Antigen—DR Isotype (mHLA–DR)

Zhuang et al. evaluated the expression of monocyte human leukocyte antigen–DR (mHLA–DR) measured within 24 h after admission in the prediction of short-term survival, and mHLA–DR levels were reported to be higher in patients with mortality when compared to survivors [45].

#### 3.3.3. C–Reactive Protein (CRP)

As an important marker of the inflammatory process, CRP has a value in the diagnosis of pneumonia to some extent. A CRP level above 40 mg/L has a sensitivity of 70% to 73%, and a specificity of 90% to 65% in diagnosing bacterial pneumonia [46]. Another study by Boussekey et al. cited that CRP had lower sensitivity when compared to PCT for the recognition of bacterial respiratory infection [47].

In the outpatient conditions, CRP levels can supply meaningful information to exclude pneumonia. In this group, the evaluation of signs and symptoms identified diagnostic risks accurately in around one-fourth of patients [48]. On the other hand, with the majority of patients in whom diagnostic doubt remained, CRP levels were useful to exclude pneumonia.

The predictive power of CAP was improved by adding biomarkers, such as CRP, to the other well-known scores. Menendez et al. reported that the added value of CRP to PSI, CURB–65, or CRB–65 augmented the prediction of death for hospitalized patients [49]. These combinations retained a sensitivity of 0.77 and a specificity of 0.78. Therefore, it can be valued as a prognostic instrument, hampered by a lack of sensitivity and/or specificity in individual decision-making. A recent meta-analytic study disclosed that CRP has been found to be the most reliable marker (AUC = 0.8), together with leukocytosis (0.77) and PCT (0.77) [50]. For CRP, LR+ and LR− were 2.08 and 0.32 (cutoff: 20 mg/L), 3.64 and 0.36 (cutoff = 50 mg/L), and 5.89 and 0.47 (cutoff: 100 mg/L), respectively. For PCT, LR+ and LR− were 2.50 and 0.39 (cutoff: 0.10 µg/L), 5.43 and 0.62 (cutoff: 0.25 µg/L), and 8.25 and 0.76 (cutoff: 0.50 µg/L), respectively. On the other hand, the combination of CRP > 49.5 mg/L with PCT > 0.1 µg/L had an LR+ of 2.24 and LR− of 0.44.

#### 3.3.4. P–Calprotectin

P–calprotectin has been recently reported to be a useful aid in sepsis-suspected patients. This biomarker has been found to be significantly elevated in critical patients after an assessment by a multidisciplinary team [51]. P–calprotectin was superior to traditional biomarkers in predicting the need for intensive care.

#### 3.3.5. Procalcitonin (PCT)

PCT, on the other hand, is a 116-amino acid precursor polypeptide for calcitonin produced in C cells of the thyroid, which is expressed in reaction to microbial toxins and pro-inflammatory mediators such as IL-1B (interleukin-1beta, TNF-α, and IL-6), bacterial products (e.g., lipopolysaccharides) and necrotic tissue cells, and immune-reactive calcitonin [52]. They act as factors to reduce serum calcium levels, and their levels can be detected in healthy adults that rapidly rise 1000-fold with severe disease states [53]. PCT also responds to modulate immunity-related functions, vasomotility, and microcirculation, as well as changes in cytokine expression during hypoperfusion states mediated by endotoxins [54]. PCT is expressed and converted to calcitonin in the C-cells in the thyroid glands of healthy people without inflammation, presenting very low PCT levels (<0.1 ng/mL) [55].

PCT is a widely used serum biomarker, which is closely related to bacterial structure and severity of the infection. It is most specific to infections incited by bacteria, as it is attenuated by INF9 expressed in response to viral infections [56]. The metabolic response to elevated PCT in critical diseases has not been explained so far. The inflammatory response is critical to understand metabolic changes during extreme stress [57].

PCT is accepted as a valuable inflammatory biomarker to discern bacterial from viral, and other causes of pneumonia [34,58]. Besides acute bacterial infections, PCT helps to identify various medical conditions, including post-surgical anastomotic leaks, acute kidney injury, and consequences of intracerebral hemorrhage [59]. Research revealed that PCT levels rise in correlation with bacteremia and severe infection and predict death in patients with CAP and sepsis [60,61]. Studies from Northern Europe pointed out a link between elevated PCT readings and pneumonia severity [62]. PCT is not routinely worked up in the diagnostic process of CAP as its predictive accuracy is only moderate. Most clinicians order a PCT level at the time of diagnosis and serially to help decide the most beneficial duration of antibiotics.

#### 3.3.6. Comparisons of PCT with CRP and Other Markers

Some investigations highlighted its diagnostic value in different clinical scenarios. PCT was more accurate than CRP to predict bacteremia, for discriminating bacterial from nonbacterial infections, and for determining bacterial species (i.e., AUC of PCT and CRP were 0.79 and 0.66, respectively) [63]. The optimal cutoff value for PCT was 0.5 mcg/L (sensitivity 70% and specificity 70%), whereas it was 50.0 mg/L for CRP (sensitivity 63% and specificity 65%). Using these cutoff values as a reference, the OR was 71.11 and the hazard ratio was 6.27 for PCT > 2.0 mcg/L, and the rate of BC positivity was markedly elevated.

Some studies advocated CRP against PCT in specific subgroups. For example, CRP was better than PCT at predicting pneumonia, as demonstrated in a retrospective study of elderly patients with comorbid diseases [64]. Zhang et al. compared patients with sepsis and those with local inflammatory diseases admitted to the ICU in China [65]. The combined AUC was significantly larger than the sum of IL–10, IL–17, and PCT. A clinical decision curve analysis disclosed that the three combined tests performed better than the individual tests with regard to the total clinical benefit rate. It was concluded that there was a considerable net therapeutic benefit ranging from 3 to 87%.

The analyses of soluble interleukin-2 receptor (sIL-2R), tumor necrosis factor-a (TNF-a), and PCT were found to carry a considerable benefit in the recognition of septic course in closed abdominal trauma complicated with severe multiple injuries [66]. The high concentrations of PCT and TNF-a can be used as valuable predictors of sepsis.

## 4. Role of PCT in Comorbidities, Special Subgroups and Associated Conditions

### 4.1. COPD and PCT

An elevated PCT level (HR: 1.02, 95% CI: 1.00–1.03) is among the variables predicting death, namely the age (hazard ratio: 1.12, 95% CI: 1.05–1.19) and a history of cancer (HR: 7.04, 95% CI: 2.22–22.36) [67]. The use of PCT-based protocols in COPD exacerbations reduced the use of antibiotics (RR: 0.56, 95% CI: 0.43–0.73) and decreased the number of days of antibiotic administration (difference in days: −3.83, 95% CI: −4.32–0.35) but had no effect on the total length of hospital stay and mortality [68].

In a study by Corti et al., patients with COPD exacerbations followed up for 28 days were randomized to PCT and control groups, and the rate of antibiotic use was more than 41% in the PCT group, whereas it was 67% in the controls [69]. Similarly, Stolz et al., reported that PCT-mediated care reduced prescriptions of antibiotics (40 versus 72%) [70].

PCT worked up in conjunction with CRP was advocated as a more useful test in the recognition and management of acute exacerbations of COPD [71]. The levels of PCT and CRP were (1.97 ± 0.13) μg/L and (7.34 ± 2.66) mg/L, respectively, in the infection group after treatment, which was significantly lower than the levels before treatment. Levels of CRP in combination with PCT is a reliable index for the detection of bacterial infection in these patients. According to the results of a retrospective cohort study conducted by Ulrich et al. in 2019, in which records of 238 COPD exacerbation cases were evaluated, the duration of antibiotic administration was not shortened in patients directed by PCT within a 6-month period, and no difference was noted in regard to mortality [72]. Meanwhile, it was reported that the rate of 30-day readmissions was reduced in the group in which PCT was measured compared to the group in which it was not worked up (21 versus 36%).

In brief, the use of biomarkers will not be sufficient alone for prognostic purposes in the management of COPD exacerbations. The use of PCT and CRP levels together with other signs and symptoms may be considered to identify the etiology of exacerbation and to predict the probability of readmission to the hospital [73]. The use of PCT-guided antibiotics in patients admitted to the hospital with COPD exacerbation does not alleviate mortality compared to standard care. Thus, routine PCT level measurement is not necessary.

### 4.2. Sepsis and PCT

Fever associated with signs of shock is commonly caused by sepsis, which encompasses a wide spectrum of illnesses caused by infection and its complications. The incidence of sepsis is increasing worldwide with a high death toll. The mortality rate of severe sepsis is between 30 and 50%, while septic shock has a death rate greater than 50% [74]. Expedient recognition and management definitely reduce mortality. On the other hand, the symptomatology and findings of a patient with sepsis are similar to other inflammatory reactions, especially when the source of infection cannot be identified. Appropriate management comprises empirical antibiotics and resuscitation. A PCT level > 2 SD above normal is a typical finding of sepsis laboratory workup [75].

CAP and severe CAP comprise a continuum in which some patients worsen and die from sepsis and septic shock. Severe sepsis can ensue at the beginning of the clinical course of the infection in more than a third of cases with CAP. Patients with severe CAP have been found to have significantly more positive BC results than those with non-severe pneumonia. ATS guidelines describe severe CAP with minor and major criteria [1]. The validated definition includes either one major criterion (septic shock and respiratory failure requiring mechanical ventilation) or three or more minor criteria (tachypnea, PaO_2_/FIO_2_ ratio < 250, multilobar infiltrates, altered mental status, uremia, leukopenia, thrombocytopenia, hypothermia, and hypotension).

### 4.3. Use of PCT Alone vs. PCT-Based Scores

Tsui et al. proposed a PCT-based score that has been revealed to perform better in detecting sepsis and compared this with PCT concentrations alone, CRP, and infection probability score [76]. The PCT-based score performed well in detecting sepsis (AUROC 0.80; 95% CI 0.74–0.85; sensitivity 0.70; specificity 0.76), which outperformed the other competitors.

### 4.4. PCT in Elderly Patients

PCT was found to be beneficial in the elderly who were investigated for bacteremia. Lee et al. pointed out that PCT was not inferior to other tests in recognition of bacterial etiology [77]. On the other hand, the deficient reliability of the test withheld recommendations on the use of the test in isolation. In a recent study on elderly patients with sepsis, PCT, IL-10, IL-6, and IL-5 were noted to be accurate in estimating ICU follow-up but were not effective in the prediction of mortality [78]. Another special population consists of those residing in nursing homes. Pneumonia and CAP are also important acquired morbidities in those dwelling in nursing homes. PCT levels were recorded to be 4.7 ± 5 ng/mL in non-survivors and 0.86 ± 1 ng/mL in survivors in this group of patients (*p* < 0.001) [79]. The AUC for PCT in estimating death was 0.84 (95% CI 0.70–0.98, *p* = 0.001). A PCT level measured as >1.1 ng/mL on presentation predicted mortality independently.

### 4.5. PCT in Congestive Heart Failure (CHF)

Elderly patients with CHF are commonly present at the ED with respiratory symptoms [13]. Some of the cases can have pneumonia-complicating CHF, and/or signs and symptoms of CAP can be confused with those of CHF. The use of biomarkers, such as PCT measurement, will help identify patients with bacterial infection and guide antibiotic therapy. Elderly patients with CHF and elevated PCT levels indicate a high probability of bacterial infection [80,81].

### 4.6. Use of PCT in Chronic Renal Insufficiency (CRI)

PCT levels were found to be greater in patients with reduced renal function as compared to the others, and levels can be lowered after dialysis by around 20–80% [82,83,84,85]. In general, PCT and CRP show poor sensitivity but adequate specificity in the recognition of bacterial infection in patients with CRI. Their negative likelihood ratio is low, which renders their value as a rule-out test questionable. In a meta-analysis published by Lu et al., the positive LR for PCT (LR)+ 6.0, 95% CI 3.1–11.4) was sufficiently high as a rule-in diagnostic tool, while the LR− was not low enough for an exclusion test (LR− 0.3, 95% CI 0.1–0.5) [86]. Therefore, clinicians are not recommended to rule out bacterial pneumonia or sepsis in a patient with normal levels of PCT.

Mean levels of serum PCT in ESRD patients on dialysis were 0.7 ng/mL, and more than a half (57%) of dialysis patients had pre-dialytic levels above 0.5 ng/mL [87]. Lee et al. measured serum PCT levels in ESRD patients on antibiotic therapy for bacterial infection (ESRD infection [iESRD] group) and compared them with those of ESRD patients on dialysis without any infection signs (ESRD control [cESRD] group [88]). Serum PCT is found to be a strong indicator of infection in ESRD patients, using 0.75 ng/mL as a cutoff value. Seventy percent of uninfected children were demonstrated to have pre-dialytic PCT levels > 0.5 ng/mL and were reduced substantially by 40% via dialysis [89].

### 4.7. PCT in Obesity

Being overweight frequently accompanies insulin resistance. PCT is reported to be an important marker of fat accumulation and metabolic parameters associated with obesity. Boursier et al. have demonstrated that PCT can be a valuable marker of fat accumulation and metabolic parameters in obese individuals [90]. PCT has also been postulated to be useful to schedule exercise and weight loss [59].

### 4.8. Use of PCT in Children

Damman et al. sought for the diagnostic accuracy of PCT in predicting bacteremia in febrile children with indwelling central lines [91]. They reported that a PCT level of ≥0.6 ng/mL was the best cutoff point, with a sensitivity of 85.6% and a specificity of 66% (AUC 0.85), which approved PCT as a sensitive biomarker predicting bacteremia in febrile children with a central line. A higher PCT was found to predict more severe pneumonia and longer hospital stays in a study that enrolled around 490 children with pneumonia [92]. The marker can be used to aid clinicians in the accurate evaluation of pneumonia severity.

Although PCT is more accurate in predicting severe infectious states when compared to CRP and other markers, it does not appear to be sensitive or specific enough to act as a reliable test for the recognition of sepsis [93]. An elevated PCT level on admission indicates a poorer outcome for patients with sepsis or septic shock, albeit serial measurements provide more reliable estimation of the clinical course.

### 4.9. H1N1 Infection (Influenza) and PCT

Most patients are diagnosed on clinical grounds alone amidst influenza epidemics. ICU patients complicated by bacterial pneumonia had higher PCT levels than those having only H1N1 infection (6 mcg/L vs. 0.6 mcg/L) [94]. The test had 80% sensitivity and 73% NPV in estimating pneumonia with the cut-off level accepted as 0.5 mcg/L. Moreover, tests had a sensitivity around 50%, but specificity was above 95% [95,96].

Wu et al. published a systematic review and meta-analysis and pointed out that the LR+ for PCT was 2.3 (95% CI: 1.9–2.8), which is not adequate for its reliability as a diagnostic tool. Meanwhile, its LR− was low enough to employ as an exclusion tool (LR− = 0.26; 95% CI: 0.1–0.4) [97]. PCT has a high sensitivity, particularly for ICU patients, but a low specificity to diagnose secondary bacterial infections for those with influenza. It can be a valuable rule-out test because of its high LR− but cannot be used as a standalone rule-in test because of suboptimal LR+.

Rodríguez et al. demonstrated that PCT has a high negative predictive value (94%), and that lower PCT readings appear to be a useful instrument to rule out coinfection, particularly for those without signs of hypoperfusion, in a prospective multicentric investigation [98]. Of note, PCT (2.4 ng/mL vs. 0.5 ng/mL, *p* < 0.001), but not CRP (25 mg/dL vs. 38 mg/dL; *p* = 0.6), was higher in patients with coinfections.

#### Secondary Bacterial Infections in Primary Influenza Virus Infection

ATS reported that bacterial pneumonia can occur concurrently with influenza virus infection or present later as a worsening of symptoms following primary influenza virus infection (1). Ten to 30% of those hospitalized for influenza and bacterial pneumonia die from these infections [99]. *S. aureus* is one of the most common bacterial infections associated with influenza pneumonia, followed by *S. pneumoniae*, *H. influenzae*, and Group A Streptococcus [100,101].

### 4.10. Use of PCT to Diagnose Bacterial Infection in Patients with Autoimmune Diseases

Pooled specificity was calculated to be 0.90 (95% CI 0.85–0.93) for PCT and 0.56 (95% CI 0.25–0.83) for CRP to identify bacterial infections in a meta-analysis [102]. The LR+ for PCT (7.3 [95% CI 5.1–10.4]) was adequate to make PCT a reliable diagnostic instrument, while the LR− (0.3 [95% CI 0.2–0.4]) was not adequately low to accept PCT as a reliable exclusion tool. In brief, PCT has a higher diagnostic reliability when compared to CRP in the diagnosis of bacterial sepsis in patients with autoimmune disease, and PCT is more specific than sensitive. A PCT test is not suggested to be used in isolation as a rule-out test.

## 5. Principles of Treatment

Empiric antibiotic therapy with a β-lactam, combined with a macrolide, respiratory fluoroquinolones, or tetracyclines, comprise a management approach commonly recommended for patients hospitalized with CAP. The prevalence of resistant microorganisms increased in recent decades, including *S. pneumoniae,* which is resistant to frequently used antibiotics (e.g., β-lactams, macrolides, and tetracyclines) [103].

The American Thoracic Society (ATS) formulated and provided the rationale for recommendations on selected diagnostic and treatment strategies for adult patients with CAP. For healthy outpatient adults without significant comorbidities or risk factors for antibiotic resistant pathogens, ATS recommended amoxicillin or doxycycline, or a macrolide for empiric treatment of CAP in adults in the outpatient setting (1). An oral administration of antibiotics is associated with a reduced risk of all-cause mortality compared with parenteral therapy based on RCTs with low-to-moderate quality [104].

For outpatient adults with serious comorbidities (e.g., chronic heart, lung, liver, or renal disease; diabetes mellitus; alcoholism; malignancy; or asplenia), ATS recommended combination therapies such as amoxicillin/clavulanate and macrolide. On the other hand, Montes–Andujar et al. have performed a network meta-analytic study recently to identify the empiric antibiotic with the highest probability of being the best (HPBB) in terms of cure rate and mortality rate in hospitalized patients with CAP [105]. They cited that for the cure rate, ceftaroline and piperaciline are the options with the HPBB. However, for the mortality rate, the treatment agents of choice are ceftriaxone plus levofloxacin, ertapenem, and amikacin plus clarithromycin. In another meta-analytic study, the efficacy of doxycycline was found to be comparable to macrolides or fluoroquinolones in mild-to-moderate CAP. Thus, it represents a viable treatment option [106]. Ashy et al. compared the clinical success attributed to specific macrolide agents and revealed that erythromycin use was associated with significantly lower rates of clinical success (RR 0.79), clinical cure (RR 0.67), and radiological success (RR 0.84) than clarithromycin [107].

The treatment success rate for azithromycin-beta-lactam after 10-to-14 days was 87.55%; for clarithromycin-beta-lactam after 5 to 7 days of therapy, it was 75.42% [108]. A shorter hospital stay was achieved with a clarithromycin-beta-lactam regimen (7.25 days versus 8.45 days). The authors recommended a macrolide and beta-lactam combination using susceptibility data from the treating facility. Five-day treatment and longer antibiotic courses for CAP resulted in similar clinical and microbiological responses and exhibited comparable safety profiles [109]. However, recent analyses pointed out that a shorter treatment duration (3–5 days) probably offers the optimal balance between efficacy and treatment burden for treating CAP in adults if they achieved clinical stability [29].

### 5.1. Use of PCT in Antibiotic Stewardship

PCT is commonly recommended to be used to exclude CAP and, thus, identify patients who do not need antibacterial therapy. In a randomized controlled trial, the use of a “PCT algorithm” helped shorten the duration of antimicrobial therapy and lowered antibiotic-related adverse effect rates [110].

The Stop Antibiotics on Procalcitonin guidance Study (SAPS) trial enrolled more than 1500 patients in the ICU with suspected or known infection (65% had a respiratory infection) [111]. Clinicians stopped antibiotics when PCT levels were below 0.5 ng/mL or decreased by ≥80% from the peak. The PCT-guided group had substantially less antibiotic use (7.5 versus 9.3 defined daily doses) and lower 28-day mortality (19% vs. 25%) than controls. Identical thresholds of ≤0.5 ng/mL, or an ≥80% reduction from the peak on Day, 5 were used in the Procalcitonin-Guided Antimicrobial Therapy to Reduce Long-Term Sequelae of Infections (PROGRESS) trial. Compared with usual care, PCT-guided antibiotic discontinuation had lower antibiotic use (5 versus 10 days) and 28-day mortality [112]. These suggestions are in line with the 2021 Surviving Sepsis Campaign and the 2016 IDSA antimicrobial stewardship guidelines for using PCT levels to guide antibiotic cessation in patients with suspected infections in the ICU [113,114].

Consensus algorithms differ with regard to the timing of treatment (i.e., initiation in low-risk patients or discontinuation in high-risk patients) and PCT cut-off points for the recommendation/strong recommendation to discontinue antibiotics (≤0.25/≤0.1 μg/L in ED and inpatients, ≤0.5/≤0.25 μg/L in ICU patients, and a reduction by ≥80% from peak levels in sepsis patients) [115]. PCT levels > 0.25 μg/L necessitate the initiation of antibiotics in those with CAP. In patients receiving antibiotics, PCT levels should be checked in around two-to-three days. Antibiotic discontinuation is considered in patients with a visible improvement, and if PCT levels are either below 0.25 μg/L or have reduced >80% from peak levels. Causes of treatment failure include empyema, multi-resistant strains, or incomplete antibiotic therapy, which should be searched in those without sufficient decrease in PCT levels.

Tonkin–Crine et al. performed a systematic review of seven studies on the results of interventions intended to influence physicians’ antibiotic prescribing behaviour for pneumonia in ambulatory care [116]. They found that PCT-guided management appears to reduce antibiotic prescribing in EDs (adjusted OR 0.3, 95% CI 0.3 to 0.4). The overall effect of these interventions was small but likely to be clinically important.

In brief, PCT-guided treatment of pneumonia and other respiratory infections helps reduce antibiotic consumption and improve clinical outcomes, although launched algorithms had differences in PCT cut-off points. If initial PCT levels are above 0.25 μg/L, a bacterial infection is unlikely, and other illnesses (e.g., pulmonary embolism or heart failure) should be ruled out.

Ventilator-associated pneumonia, acute bronchitis, and acute exacerbations of COPD, as well as other chronic lung diseases, do not routinely require PCT levels for management. PCT work-up can be reserved for selected cases in these situations.

### 5.2. Value of PCT in Outpatient Follow-Up

PCT can be ordered in combination with a positivity for influenza or COVID-19 to withhold antibiotics or stop them early, provided that there are no other signs of bacterial infection (e.g., dense consolidation on chest imaging, increased WBC count, or positive BC) and that the patient has a close follow-up. Other experts prefer to treat empirically with antibiotics because bacterial CAP is hard to exclude definitively, the morbidity association with CAP is high, and outpatient treatment courses for CAP are short (e.g., five days).

While data directly supporting PCT-guided antibiotic decision-making in outpatients with CAP are limited, one randomized trial comparing >150 outpatients with CAP found that clinical outcomes were similar, and that antibiotic exposure was decreased when PCT was used [117]. A recent study evaluated 469 general practice patients with any kind of respiratory infections [118]. Antibiotic use was reduced at a rate of 26% when PCT was used to guide antibiotic decision-making, while clinical outcomes were similar. Interestingly, point-of-care lung USG did not further reduce antibiotic prescription, although a potential added value cannot be excluded.

### 5.3. General Trends in Procalcitonin (PCT) Levels with Agents and Clinical Conditions

Table 3 summarizes and lists the conditions triggering increases in PCT levels with and without pneumonia. False positive and negative PCT elevations should be accounted for in clinical grounds before affecting decision-making processes [119].

## 6. Blank Areas for PCT Use

PCT had shortcomings that restricted its use in most decision-making processes. Severe inflammatory states included major surgery, trauma, burns, inhalational injury, necrotic lesions of pancreas, and hypothermic treatment methods [120,121,122].

## 7. Conclusions

Pneumonia and sepsis are diseases of utmost importance to diagnose and treat urgently, since they are responsible for a high worldwide death toll. Although an elaborate history and clinical examination are mainstays for an accurate diagnostic process, certain laboratory adjuncts can comprise invaluable aids in appropriate management. PCT is one of the extensively investigated sepsis biomarkers also employed in pneumonia in a wide range of studies, together with CRP and a complete blood count. It appears to be a reliable diagnostic tool to expedite care of these patients with severe infections in the acute setting.

PCT is an inflammatory biologic marker that can be useful in distinguishing between bacterial and nonbacterial etiologies of pulmonary infection. It was reported to be more accurate than most other acute phase reactants and indicators as a predictor of pneumonia, bacteremia, sepsis, and poor outcomes. In addition, PCT use is beneficial to judge timing to stop antibiotic therapy in serious infections. The physicians need to gauge pluses and minuses of the useful biomarkers in expedient recognition and management of severe infections.

PCT can help direct antibiotic decisions for the treatment of acute respiratory infections to minimize antibiotic prescription and orders to improve prognoses in this regard. PCT algorithms may be adapted to the type of infection and the unique case scenario. PCT has shown potential benefits in antibiotic stewardship protocols.

## Figures and Tables

**Table 1 diagnostics-13-01869-t001:** Frequency of etiologic agents of community-acquired pneumonia (CAP) in elderly patients.

*Streptococcus pneumoniae*	Up to 50%
Atypical (*Legionella pneumophila* and others)	Up to 25%
*Haemophilus influenzae*	0–13%
*Staphylococcus aureus*	0–7%
Methicillin-resistant *Staphylococcus aureus* (MRSA)	0–6%
Gram-negative bacilli including *Pseudomonas aeruginosa*	Up to 27%
Virus	0–8%
*Aspiration pneumonia*	10%

**Table 2 diagnostics-13-01869-t002:** Differential diagnosis of patients with cough. These entities commonly masquerade as CAP.

*Pulmonary embolism*
Cryptogenic organizing pneumonia
*Tuberculosis*, *Actinomycosis*
*Pulmonary vasculitis*, lupus pneumonitis and hypersensitivity pneumonitis, acute or chronic eosinophilic pneumonia
Sickle cell syndrome, sickling crisis
Acute hemorrhage in the alveoli
*Radiation pneumonitis*
Leukemia and neoplasms such as bronchogenic carcinoma
Drug-induced pulmonary infiltration

**Table 3 diagnostics-13-01869-t003:** Procalcitonin (PCT) levels with regard to various microbiologic agents and clinical variables.

	PCT > 0.25 ng/mL	PCT < 0.25 ng/mL
Bacterial infections (PCT levels may not increase in case of abscesses or empyema)		
Typical pneumonia agents	Most reported thus far	
Atypical pneumonia agents	*Legionella* spp., *mycobacteria* spp.	*Chlamydia pneumonia,* *Mycoplasma pneumonia,* *mycobacteria* spp.
Viral agents	None	All
Fungal agents	*Candida* spp.	*Mucormycosis,* *aspergillosis,* *coccidioidomycosis*
Parasitic agent	*Plasmodium* spp. (malaria)	
Physiologic stress	Trauma, surgery, burn, pancreatitis Bowel ischemia Cerebrovascular accident, intracerebral hemorrhage Shock (all causes)	
Toxin-mediated entities	*Clostridioides difficile*-associated disease, mushroom poisoning	*C. difficile* colonization
Comorbid diseases, immune and rheumatologic entities	Kawasaki disease, renal and hepatic failure	*Rheumatoid arthritis,* gout and pseudogout, Behçet disease Crohn’s disease Systemic lupus erythematosus, polyarteritis nodosa, temporal arteritis, granulomatosis with polyangiitis
Malignant tumors	Thyroid and lung cancers	Lymphoma and sarcoma Splanchnic cancers (e.g., pancreatic and renal cell carcinoma)

## Data Availability

Data sharing not applicable. No new data were created or analyzed in this study. Data sharing is not applicable to this article.

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
