# Peer review of "Diagnostic and Prognostic Roles of Procalcitonin and Other Tools in Community-Acquired Pneumonia: A Narrative Review"

_diagnostics, 2023, doi:10.3390/diagnostics13111869_

Round 1

Reviewer 1 Report

Introduction How much is the precent of CAP in Turkey, % of bacteria….

On the other hand, elderly patients have a different order of frequency of culprit agents in CAP (Table 1) (10,16, 17) literature from 2009, 2017 and 2001

Principles of treatment

Empiric antibiotic therapy with a β-lactam combined with a macrolide, respiratory fluoroquinolones, or tetracyclines comprise a management approach commonly recommended for patients hospitalized with CAP. Prevalence of resistant microorganisms increased in the recent decades including S. pneumoniae that is resistant to frequently used 348 antibiotics, (e.g., β-lactams, macrolides, and tetracyclines) (99).  – literature is from 2014 ….. There is Guidelines from 2019 from ATS

(American Journal of Respiratory and Critical Care Medicine, Volume 200, Issue 7, 1 October 2019)

Literature 2023-2018 – 18.7 %, 2017-2013 – 35.7 % (last ten years 54.4 %), under 2012 45.6 %.

The study must be improved with newest literature because this is “Review” paper.

Author Response

Response to the reviewers,

Dear Editors and referees,

Thank you for your valuable comments. We have worked on these extensively and accomplished the necessary changes.

Cordially yours,

Ozgur KARCIOGLU, M.D., Prof, FEMAT

Dept. of Emergency Medicine,
Univ. of Health Sciences, Taksim Training and Research Hospital, 

Beyoglu, Istanbul

Email: okarcioglu@gmail.com

Phone: +90.505.5252399 

Rev. #1:

  1. Introduction How much is the precent of CAP in Turkey, % of bacteria….The following sentences were added:

Different studies revealed various prevalence figures in pneumonia. The annual incidence of pneumonia is reported to vary between 0.28 and 1.16%, and both incidence and mortality increase especially in advanced ages (Marrie, Almirall).

Similar to the universal data, the incidence of pneumonia increases with age in Turkey. Pneumonia mortality varies between 1-60% in relation to the severity of the disease, and this rate rises significantly in hospitalized patients (10.3-60%) (Köksal, Kurutepe). Patients with pneumonia accounted for 1.9% of all hospitalizations in Turkey (Bulbul).

In a population-based multicentric study in Turkey, diagnosis of PP was based on the presence of at least one positive laboratory test result for Streptococcus pneumoniae (blood culture or sputum culture or urinary antigen test) in patients with radiographic findings of pneumonia. 465 patients were diagnosed with CAP, of whom 59 (12.7%) had pneumococcal pneumonia (Şenol). In another study focusing on elderly patients in Turkey, Pseudomonas spp. (n = 8, 26.6%) was the most common agent, followed by Streptococcus pneumoniae (n = 7, 23.3%) (Surme). Other typical bacterial agents were Staphylococcus aureus (13.3%), Haemophilus influenzae (10%) and Acinetobacter spp. (10.0%). Around one-fourth of the Staph. aureus strains had methicillin resistance.

  1. On the other hand, elderly patients have a different order of frequency of culprit agents in CAP (Table 1) (10,16, 17) literature from 2009, 2017 and 2001 The response to the previous item (1) includes enrichment of this information also. 

  2. Principles of treatment

Empiric antibiotic therapy with a β-lactam combined with a macrolide, respiratory fluoroquinolones, or tetracyclines comprise a management approach commonly recommended for patients hospitalized with CAP. Prevalence of resistant microorganisms increased in the recent decades including S. pneumoniae that is resistant to frequently used 348 antibiotics, (e.g., β-lactams, macrolides, and tetracyclines) (99).  – literature is from 2014 ….. There is Guidelines from 2019 from ATS

(American Journal of Respiratory and Critical Care Medicine, Volume 200, Issue 7, 1 October 2019)

Response:

The section was supported by more recent data:

“The American Thoracic Society (ATS) panel formulated and provided the rationale for recommendations on selected diagnostic and treatment strategies for adult patients with CAP. For healthy outpatient adults without significant comorbidities or risk factors for antibiotic resistant pathogens, ATS recommended amoxicillin or doxycycline, or a macrolide for empiric treatment of CAP in adults in the outpatient setting (Metlay19).

For outpatient adults with serious comorbidities (e.g., chronic heart, lung, liver, or renal disease; diabetes mellitus; alcoholism; malignancy; or asplenia) ATS recommended combination therapies such as amoxicillin/clavulanate and macrolide.

p.11:  Secondary bacterial infections in primary influenza virus infection: ATS reported that bacterial pneumonia can occur concurrently with influenza virus infection or present later as a worsening of symptoms following primary influenza virus infection (Metlay19). 10% to 30% of those hospitalized for influenza and bacterial pneumonia die from these infections (Metersky12). S. aureus is one of the most common bacterial infections associated with influenza pneumonia, followed by S. pneumoniae, H. influenzae, and group A Streptococcus (Jean 10, Paddock 12).

Literature 2023-2018 – 18.7 %, 2017-2013 – 35.7 % (last ten years 54.4 %), under 2012 45.6 %.

The study must be improved with newest literature because this is “Review” paper.

Response: A number of references was replaced by more recent data which were highlighted with blue in the manuscript references section.

Reviewer 2 Report

This review focuses on CAP so the title should say CAP rather than pneumonia. In addition, the review not only discuss the roles of procalcitonin, but also covers other diagnostic tools of CAP, and the title needs to reflect the scope.

3.10.2 and 3.10.3 are complications of CAP, authors need to introduce the prevalence and impact on mortality of these conditions.

3.10.5 through 3.10.13 are special patient populations to consider when evaluating PCT result, these sections need to be better organized, either having a subtitle at the beginning of these sections or to combine them into fewer sections.

There needs to be discussions on the variable PCT results from different analyzers and the cutoffs of PCT in infants, children and adults.

The paragraphs are shorter than usually expected.

Author Response

Response to the reviewers,

Dear Editors and referees,

Thank you for your valuable comments. We have worked on these extensively and accomplished the necessary changes.

Cordially yours,

Ozgur KARCIOGLU, M.D., Prof, FEMAT

Dept. of Emergency Medicine,
Univ. of Health Sciences, Taksim Training and Research Hospital,

Beyoglu, Istanbul

Email: okarcioglu@gmail.com

Phone: +90.505.5252399 

Rev. #2

Comments and Suggestions for Authors

This review focuses on CAP so the title should say CAP rather than pneumonia. In addition, the review not only discuss the roles of procalcitonin, but also covers other diagnostic tools of CAP, and the title needs to reflect the scope.

Response: The title was revised to cover the article’s contents as follows: Diagnostic and prognostic roles of procalcitonin and other tools in community-acquired pneumonia: A narrative review

Running head: Procalcitonin and other diagnostic tools in pneumonia

3.10.2 and 3.10.3 are complications of CAP, authors need to introduce the prevalence and impact on mortality of these conditions.

Response: Sepsis section was enlarged with explanations (p.10). 

3.10.5 through 3.10.13 are special patient populations to consider when evaluating PCT result, these sections need to be better organized, either having a subtitle at the beginning of these sections or to combine them into fewer sections.

Response: These sections have been enumerated and re-organized.

There needs to be discussions on the variable PCT results from different analyzers and the cutoffs of PCT in infants, children and adults.

Response: Infants and children are beyond the scope of this article.

Comments on the Quality of English Language

The paragraphs are shorter than usually expected.

Response: Most paragraphs were re-organized and merged.

Round 2

Reviewer 1 Report

The quality of work with new data after revision has improved.

Reviewer 2 Report

Authors have addressed the reviewers' comments.